# Bixin Combined with Metformin Ameliorates Insulin Resistance and Antioxidant Defenses in Obese Mice

**DOI:** 10.3390/ph17091202

**Published:** 2024-09-12

**Authors:** Camila Graça Pinheiro, Bruno Pereira Motta, Juliana Oriel Oliveira, Felipe Nunes Cardoso, Ingrid Delbone Figueiredo, Rachel Temperani Amaral Machado, Patrícia Bento da Silva, Marlus Chorilli, Iguatemy Lourenço Brunetti, Amanda Martins Baviera

**Affiliations:** 1Department of Clinical Analysis, School of Pharmaceutical Sciences, São Paulo State University (UNESP), Araraquara 14800-903, SP, Brazil; camila.pinheiro@unesp.br (C.G.P.); motta.bp@gmail.com (B.P.M.); juliana.oriel@gmail.com (J.O.O.); felipe.n.cardoso@unesp.br (F.N.C.); ingrid.delbone@unesp.br (I.D.F.); iguatemy.brunetti@unesp.br (I.L.B.); 2Department of Drugs and Medicines, School of Pharmaceutical Sciences, São Paulo State University (UNESP), Araraquara 14800-903, SP, Brazil; racheltemperani@gmail.com (R.T.A.M.); patrbent@gmail.com (P.B.d.S.); marlus.chorilli@unesp.br (M.C.)

**Keywords:** carotenoids, advanced glycation end products, glycoxidative stress, antioxidant enzymes, natural bioactives

## Abstract

Bixin (C_25_H_30_O_4_; 394.51 g/mol) is the main apocarotenoid found in annatto seeds. It has a 25-carbon open chain structure with a methyl ester group and carboxylic acid. Bixin increases the expression of antioxidant enzymes, which may be interesting for counteracting oxidative stress. This study investigated whether bixin-rich annatto extract combined with metformin was able to improve the disturbances observed in high-fat diet (HFD)-induced obesity in mice, with an emphasis on markers of oxidative damage and antioxidant defenses. HFD-fed mice were treated for 8 weeks with metformin (50 mg/kg) plus bixin-rich annatto extract (5.5 and 11 mg/kg). This study assessed glucose tolerance, insulin sensitivity, lipid profile and paraoxonase 1 (PON-1) activity in plasma, fluorescent AGEs (advanced glycation end products), TBARSs (thiobarbituric acid-reactive substances), and the activities of superoxide dismutase (SOD), catalase (CAT), and glutathione peroxidase (GSH-Px) in the liver and kidneys. Treatment with bixin plus metformin decreased body weight gain, improved insulin sensitivity, and decreased AGEs and TBARSs in the plasma, liver, and kidneys. Bixin plus metformin increased the activities of PON-1, SOD, CAT, and GSH-Px. Bixin combined with metformin improved the endogenous antioxidant defenses in the obese mice, showing that this combined therapy may have the potential to contrast the metabolic complications resulting from oxidative stress.

## 1. Introduction

Obesity and overweight are highly prevalent clinical conditions and act as risk factors for the development of type 2 diabetes mellitus (T2DM). These conditions trigger cellular processes responsible for the development of insulin resistance, culminating in chronic hyperglycemia typical of DM [1]. On the other hand, chronic hyperglycemia contributes to the establishment of glycoxidative stress, which is responsible for tissue damage, induced by the glycation of macromolecules and the excessive production of advanced glycation end products (AGEs) and reactive oxygen species (ROS) [2]. Therefore, the adoption of therapeutic regimens that can reduce both glycemia and glycoxidative stress becomes important, aiming to mitigate the complications of these metabolic diseases [3].

Metformin is the most used drug for the treatment of T2DM, as it acts efficiently in glycemic control, mainly through the inhibition of hepatic gluconeogenesis [4]. However, despite its good efficiency in controlling glycemia, metformin therapy can cause important adverse effects, which are responsible for treatment abandonment in up to 10% of cases [5]. Furthermore, the glycemic control achieved with the use of metformin is not able to prevent the development of long-term complications of T2DM given its mechanisms related to metabolic memory, which is defined as the persistence of complications caused by hyperglycemia even after the achievement of glycemic control due to several detrimental changes triggered by AGEs and ROS [6].

Bioactive compounds of natural origin become interesting in the search for complementary options to the traditional therapy for T2DM, as many of them have antioxidant and antiglycation activities. Among these interesting natural bioactive compounds are carotenoids. Carotenoids have a chemical structure that allows the accommodation of unpaired electrons from radical species, which makes them potent antioxidants. Bixin (C_25_H_30_O_4_; 394.51 g/mol) is the main apocarotenoid (70–80%) constituent of seeds from annatto (*Bixa orellana* L.), which grows in tropical countries from Central and South America to India, Indonesia, and East Africa. Annatto is widely used in popular cuisine as a condiment. Bixin has a 25-carbon open chain structure with double bonds and contains a methyl ester group and carboxylic acid. It is a carotenoid pigment with reddish-orange tonality. Bixin and annatto extracts are known for their antioxidant properties and beneficial effects on the prevention and treatment of various diseases [7,8]. Bixin can interact with important transcription factors to promote effective cellular responses in controlling lipid metabolism and combating oxidative stress, including peroxisome proliferator-activated receptor γ (PPARγ) and nuclear factor erythroid 2-related factor 2 (NRF-2) [9,10].

It seems interesting to explore combined therapy strategies based on the use of a traditional antidiabetic drug (such as metformin) and a natural bioactive compound (such as bixin), aiming to achieve responses related to glycemic control and mitigating the mechanisms associated with metabolic memory and glycoxidative stress. However, both metformin and bixin have low bioavailability [11,12], which makes it difficult to promote their pharmacological effects in vivo. This situation can be overcome by incorporating bixin into pharmaceutical delivery systems, including nanostructured lipid systems, which allow greater interaction with the aqueous environment and biological membranes, increasing absorption and reducing the need for high doses of drugs during oral administration [13,14].

The proposal of this study was to investigate the impacts of a bixin-rich annatto extract, alone or co-administered with metformin and delivered in a nanostructured lipid system, on physiometabolic parameters, glycoxidative stress biomarkers, and antioxidant defenses in an in vivo model system of obesity and insulin resistance.

## 2. Results

### 2.1. Physiological and Biochemical Parameters

From the fifth week of the experiment, the mice fed an HFD showed a significant increase in their body weight gain in comparison with the corresponding values of the C group. From the 12th week, the HFD-fed mice belonging to the M, B1, B2, MB1, and MB2 groups had low body weight gain in comparison with the mice from the H group (Figure 1).

As expected, the H mice exhibited decreased food intake, accompanied by increased energy intake. Treatments with bixin-rich annatto extract and metformin, alone or in combination, were not able to promote changes in either the food or energy intake of the HFD-fed mice until the end of the treatments (Figure 2).

There were significant increases in the weights of the epididymal white adipose tissues of the H and V mice in relation to these corresponding values for the C mice. This finding, together with the increase in body weight gain, indicates that the production of an in vivo model system of obesity occurred in the mice fed an HFD. Treatments of the HFD-fed mice with metformin or bixin-rich annatto extract (both doses), alone or in combination, decreased the weights of their epididymal white adipose tissues in comparison with the corresponding values for the H mice (Table 1). Treatments with metformin or 5.5 and 11 mg/kg bixin-rich annatto extract, alone or in combination, also decreased the weights of their brown adipose tissues in comparison with the values from the C group (Table 1). The weights of the livers of the mice treated with metformin + 5.5 mg/kg bixin-rich extract also decreased when compared with the values found in the mice from the H group (Table 1).

No significant differences were observed in the weights of the kidneys, hearts, or gastrocnemius skeletal muscles between the experimental groups studied (Table 1).

The mice from the H and V groups had high plasma levels of total cholesterol and high-density lipoprotein cholesterol (HDL-cholesterol) and low levels of triglycerides when compared to the values found in the mice from the C group. The mice from the M, B1, B2, MB1, and MB2 groups had low levels of triglycerides when compared to the mice from the C group, without differences in comparison to the values found in the mice from the H and V groups. The plasma levels of cholesterol were significantly decreased in the mice from the M, B1, B2, MB1, and MB2 groups in comparison to the values of the H group, while the HDL-cholesterol levels were decreased in the mice from the M, MB1, and MB2 groups when compared to the corresponding values of the H group (Table 2).

No changes were observed in the plasma levels of creatinine, uric acid, albumin, or ALT among the groups (Table 2).

### 2.2. Glucose Tolerance and Insulin Sensitivity

In the oral glucose tolerance test (OGTT), 15 min after glucose overload, the mice from the H and V groups had higher glycemic peaks than the values found in the mice from the C group. After 120 min, the mice from the H and V groups continued to have elevated glycemic values when compared to the corresponding values for the C group (Figure 3A,C). Therefore, glucose intolerance occurred in the mice from the H and V groups (Figure 3B,D).

The HFD-mice treated with metformin showed improvements in their glucose tolerance (Figure 3B,D). The mice from the M group exhibited low fasting glycemia, a minor glycemic peak after the glucose overload, and low glycemia levels after 120 min in the OGTT when compared to the corresponding values in the H and V groups (Figure 3A,C). Treatment with the bixin-rich annatto extract, alone or in combination with metformin, did not improve the glucose tolerance of the HFD-fed mice (Figure 3).

In the insulin tolerance test (ITT), after the administration of insulin, the mice from the H and V groups showed high glycemia levels throughout the monitoring period in comparison with the corresponding values for the C mice (Figure 4A,C). This finding supports the decreased insulin sensitivity in the mice from the H and V groups (Figure 3B,D).

Treatment with metformin or bixin-rich annatto extract (5.5 and 11 mg/kg) alone was able to improve the insulin sensitivity in the HFD-fed mice (Figure 4). The combined therapies between metformin and bixin-rich annatto extract, at both doses tested, improved insulin sensitivity as efficiently as the treatments with the bioactives alone, which suggests that the combined therapies maintained the beneficial effects of the isolated therapies on the insulin sensitivity of the HFD-fed mice.

### 2.3. Biomarkers of Glycoxidative Stress and Antioxidant Defenses in Plasma

The mice from the H group showed significantly high plasma levels of fluorescent AGEs (Figure 5A) and TBARSs (Figure 5B) when compared to the corresponding values for the C group. The V group also had high plasma levels of AGEs (Figure 5A) and TBARSs (Figure 5B) at similar values to those for the H group. Therefore, the establishment of glycoxidative stress in the H and V groups occurred as a probable consequence of glucose intolerance and chronic hyperglycemia.

There were significant decreases in the levels of fluorescent AGEs (Figure 5A) and TBARSs (Figure 5B) in the plasma of the M, B1, MB1, B2, and MB2 mice when compared to the corresponding values found in the H mice, without differences between treatments, suggesting that metformin and 5.5 and 11 mg/kg bixin-rich annatto extract decreased glycoxidative stress in the plasma of the HFD-fed mice. 

The activity of the antioxidant enzyme paraoxonase-1 (PON-1) in the plasma of the H and V mice was significantly lower than values found in the C mice (Figure 5C).

Treatment with the lower dose (5.5 mg/kg) of bixin-rich annatto extract was not able to improve PON-1 activity. On the other hand, the HFD-fed mice treated with metformin or with bixin-rich annatto extract at the higher dose (11 mg/kg) recovered in their PON-1 activity, the levels of which were similar to those of the C group. About the combined therapies, treatment with metformin + 11 mg/kg bixin-rich annatto extract increased PON-1 activity in a similar way to the isolated therapies. On the other hand, the increase in PON-1 activity observed in the HFD-fed mice treated with metformin + 5.5 mg/kg bixin-rich annatto extract was higher than that based on the effects of bixin alone, suggesting that the combined therapy had an additive effect on triggering endogenous antioxidant defenses (Figure 5C).

### 2.4. Biomarkers of Glycoxidative Stress and Antioxidant Defenses in the Liver

The levels of fluorescent AGEs (Figure 6A) and TBARSs (Figure 6B) in the livers of the HFD-fed mice from the H and V groups were higher than the values found in the C group. In parallel, the activities of the antioxidant enzymes SOD (Figure 6C), CAT (Figure 6D), and GSH-Px (Figure 6E) were decreased in the H and V mice. These findings support the establishment of oxidative stress in the livers of mice fed an HFD.

The treatments with metformin and bixin-rich annatto extract (5.5 and 11 mg/kg), alone or in combination, significantly decreased the hepatic levels of AGEs (Figure 6A) and TBARSs (Figure 6B) in comparison with the values found in the H and V mice. In addition, the treatments with metformin and bixin-rich annatto extract at both doses, alone or in combination, increased the activity of CAT in the livers of the HFD-fed mice (Figure 6D). Conversely, when metformin and bixin-rich annatto extract (5.5 and 11 mg/kg) were administered alone, these bioactives did not improve the activities of SOD (Figure 6C) or GSH-Px (Figure 6E). On the other hand, the HFD-fed mice treated with metformin + bixin-rich annatto extract, at both doses, had notable increases in their activities of SOD (Figure 6C) and GSH-Px (Figure 6E), suggesting that the combined therapy had synergistic effects on triggering endogenous antioxidant defenses in the livers of these HFD-fed mice.

### 2.5. Biomarkers of Glycoxidative Stress and Antioxidant Defenses in the Kidneys

As well as their plasma and livers, the onset of oxidative stress was also observed in the kidneys of the HFD-fed mice. Increased levels of fluorescent AGEs (Figure 7A) and TBARSs (Figure 7B) were found in the kidneys of the HFD-fed mice from the H and V groups, whose values were higher than those in the C group. In addition, the activities of the antioxidant enzymes SOD (Figure 7C) and CAT (Figure 7D) were decreased in the kidneys of the H and V mice.

The levels of AGEs (Figure 7A) and TBARSs (Figure 7B) were significantly decreased in the kidneys of the mice treated with metformin and bixin-rich annatto extract (5.5 and 11 mg/kg), alone or in combination, when compared with the values found in the H and V mice. However, it is interesting to note that metformin did not improve the activities of the antioxidant enzymes SOD (Figure 7C), CAT (Figure 7D), or GSH-Px (Figure 7E).

Bixin-rich annatto extract, at both doses, alone or in combination with metformin, caused significant increases in the activities of SOD (Figure 7C), CAT (Figure 7D), and GSH-Px (Figure 7E) in the kidneys of the mice fed an HFD, whose effects were greater than those of metformin and maintained the antioxidant responses of bixin-rich annatto extract, which reinforces that the combined therapy had beneficial effects on triggering their endogenous antioxidant defenses.

## 3. Discussion

The present study provides evidence regarding the effects of a therapeutic strategy based on combining metformin and a bixin-rich annatto extract to combat metabolic disturbances. The main beneficial effects promoted by this combined therapy were the following: (i) bixin-rich annatto extract + metformin had antiobesogenic effects; (ii) bixin-rich annatto extract + metformin improved the insulin sensitivity of the HFD-fed mice; and (iii) bixin-rich annatto extract + metformin had the ability to trigger their endogenous antioxidant machinery. 

The increase in the body weight gain and the accumulation of fat in the white adipose tissues of the mice maintained on an HFD occurred as expected in this in vivo model system of obesity. Before the beginning of the treatments (from week 1 to week 8), the HFD-fed mice showed higher body weight values than the corresponding values in the mice fed the control (C) diet, denoting the production of obesity. After the beginning of the treatments with metformin or bixin-rich annatto extract at 5.5 and 11 mg/kg, alone or in combination, the HFD-fed mice had decreases in their body weight gain when compared to untreated the HFD-fed mice (the H group). Furthermore, metformin and/or bixin-rich annatto extract partially prevented fat accumulation in their white adipose tissues. Therefore, an antiobesogenic effect can be attributed to the treatments with metformin and bixin-rich annatto extract, alone or in combination, without changes in food or energy intake. Previously, a study by Gutierrez and Romero [15] observed that the treatment of HFD-fed mice with bixin at doses of 5 and 10 mg/kg for 14 weeks also decreased their body weight gain without decreasing their food intake.

The establishment of obesity favors an increase in the concentrations of free fatty acids and circulating triglycerides, as well as hyperglycemia due to the development of insulin resistance, which is related to lipotoxicity, occurring in animals with obesity [16,17]. These changes favor, at least in part, an increase in the levels of ROS in the intracellular environment and subsequent mitochondrial dysfunction, with losses in β-oxidation and increases in the production of fatty acid radicals in the mitochondrial matrix, leading to events such as the uncoupling of electron transport chains, increased production of superoxide radical anions, and the establishment of oxidative stress [18]. 

Excess free fatty acids in the cytoplasm serve as a source of precursors for the synthesis of molecules such as diacylglycerol and ceramides, which participate in the development of insulin resistance, culminating in the inhibition of the translocation of vesicles containing the glucose transporter type 4 (GLUT 4) to the plasma membrane of myocytes and adipocytes, thus reducing glucose uptake and contributing to an increase in blood glucose levels [19]. This process is directly related to the results of the OGTT and ITT assays, which point to the establishment of glucose intolerance and insulin resistance, respectively, in the mice from the H and V groups. 

The mice from the H and V groups had increased fasting glycemia levels in relation to the values of the mice fed the C diet. The mice fed an HFD and treated with metformin and bixin-rich annatto extract, alone or combination, had better glycemic control when compared to the mice from the H group. Therefore, there is favorable potential for these treatments since metformin + bixin-rich annatto extract improved insulin sensitivity in the animals with obesity.

Glucose intolerance and chronic hyperglycemia contribute to the onset of glycoxidative stress, characterized by an exacerbation of the generation of AGEs in tissues where the uptake of glucose occurs in a non-insulin-dependent manner, including the liver and kidneys [20]. The antihyperglycemic effects of metformin and bixin-rich annatto extract may have contributed to the decrease observed in the levels of fluorescent AGEs in the livers and kidneys of the mice fed an HFD, where the effects of the treatments were most evident. It is known that AGEs are capable of activating signaling pathways that culminate in the activation of kinases such as IKKβ (inhibitory kappa B kinase beta) and JNK (c-Jun N-terminal kinase), both related to the inhibitory phosphorylation of IRS-1 (insulin receptor substrate 1), leading to reduced insulin sensitivity. Apocarotenoids, including bixin, are capable of inhibiting the signaling pathways that activate IKKβ and JNK, contributing to the restoration of insulin sensitivity, which may be related to improvements in insulin sensitivity [21,22]. On the other hand, there are studies showing that the antihyperglycemic activity of bixin may be associated with its ability to reduce carbohydrate absorption via the inhibition of the enzymes α-glucosidase and α-amylase [15]. Additionally, the possibility that bixin may have improved the pancreatic function in the mice fed an HFD cannot be ruled out. Although there are no studies on the effects of bixin or other carotenoids acting as inhibitors of dipeptidyl peptidase-4 (DPP-4), there is growing evidence on bioactives from natural sources having antidiabetic activity via DPP-4 inhibition [23,24]. Furthermore, there is great encouragement of studies on bioactives of a natural origin that act as DPP-4 inhibitors since they possess significant antioxidant properties and their use may be an effective strategy for overcoming oxidative stress in pancreatic β-cells and other important tissues, in parallel to treating diabetes [25]. In this sense, considering the promising results of this study regarding the significant potential of bixin to trigger endogenous antioxidant defenses, this set of information greatly encourages future studies on the mechanism of the antihyperglycemic activity of bixin and its relation to the inhibition of DPP-4.

Metformin has low bioavailability, reaching an absorption rate between 40 and 60% of the administered dose (peak plasma concentration). This is explained by metformin being in its protonated form at a physiological pH, which makes it difficult for it to diffuse through biological membranes. In this context, metformin depends on membrane transporters to enter cells and promote its pharmacological effects [26]. Thus, it is necessary to administer high doses of metformin, and consequently, this increases the chances of adverse effects occurring. The use of a nanostructured lipid system makes it possible to overcome the problem of the low bioavailability of metformin since encapsulation allows greater absorption of the metformin carried in the structure of the vesicles that form the nanostructured system [27]. It can be assumed that once it is incorporated into a nanoemulsion, metformin does not primarily require the process to be mediated by the transporters present in enterocytes to access the cellular environment. Consequently, a low dose of metformin would be required to obtain its therapeutic effect.

The mice fed an HFD and treated with metformin showed low fasting glycemia values, improvements in their glucose tolerance, and increased insulin sensitivity when treated with a dose of 50 mg/kg of the drug, a lower dose than those used in studies with the same in vivo model system of obesity but without the use of a nanostructured lipid system [28,29].

A nanostructured lipid system can also be used to deliver bixin due to this system’s ability to increase the absorption of this apocarotenoid, also carried in the micelle’s structure and integrated into the lipid portion of the vesicle. Due to the amphipathic nature of the nanostructured lipid system, the interaction of bixin with the aqueous medium can be facilitated, increasing its absorption [30,31]. This conformation allows more efficient delivery of bixin, enabling its more evident antioxidant potential in the plasma, liver, and kidneys, as well as promoting improvements in insulin sensitivity.

After offering an HFD for 17 weeks, increases in the plasma levels of total cholesterol and HDL-cholesterol and a decrease in the levels of triglycerides were observed, reproducing previous findings of our laboratory [32]. This decrease in the plasma levels of triglycerides may be related to an increase in the tissue’s accumulation of this lipid promoted by the onset of obesity, observed mainly in the increases in the weight of the white adipose tissues of the animals maintained on the HFD.

The enzyme PON-1 is associated with HDL in plasma, and its activity is related to the protection of this lipoprotein and others, including low-density lipoprotein (LDL), from oxidation [33]. The PON-1 activity was significantly decreased in the plasma of the H and V mice. It is known that in a scenario of hyperglycemia, a decrease in PON-1 activity may be related to the occurrence of the glycation of this enzyme [34,35], which may explain, at least in part, the low PON-1 activity in the plasma of the mice from the H and V groups. In a previous study from our laboratory, we observed that treatment with 5.5 mg/kg of bixin-rich annatto extract for 50 days was also able to increase the PON-1 activity in the plasma of streptozotocin-induced diabetic rats. [36]. The increased PON-1 activities in the plasma of the HFD-fed mice treated with metformin, bixin-rich annatto extract at the highest dose, and combinations of metformin + bixin-rich annatto extract (the MB1 and MB2 groups) may be related to improvements observed in their glycemic control, culminating in less glycation of this enzyme [37]. 

The enzymes analyzed in this study are part of the first-line antioxidant defenses. Superoxide dismutase (SOD) is the first enzyme in ROS detoxification and the most powerful antioxidant enzyme in the cellular environment. Its activity consists of the dismutation of two superoxide radical anion molecules into hydrogen peroxide and molecular oxygen, thus reducing the damage potential of this reactive species. Catalase (CAT) acts on hydrogen peroxide molecules, reducing them into water and molecular oxygen and using iron and manganese ions as cofactors. The enzyme glutathione peroxidase (GSH-Px) acts on hydrogen peroxide, reducing it into water, but also catalyzes the reduction of lipoperoxides into their corresponding alcohols, mainly in the mitochondria [38,39].

Carotenoids and apocarotenoids, including bixin, are involved in the modulating pathways that activate NRF-2, a transcription factor that acts as a positive regulator of responses to oxidative stress [40]. Therefore, when activating NRF-2, there may be an increase in the expression of these first-line antioxidant enzymes [40,41], which may explain the results found in this study for the three antioxidant enzymes analyzed in the livers and kidneys of the HFD-fed mice treated with bixin-rich annatto extract. Previously, Assis and collaborators [36] also found increases in the activities of SOD and CAT in the livers of streptozotocin-induced diabetic rats treated with 5.5 mg/kg of bixin-rich annatto extract for 50 days.

With increases in the activities of antioxidant enzymes in the mice fed an HFD and treated with metformin or bixin-rich annatto extract, a decrease in the oxidative stress induced by obesity is expected. This was observed in the analysis of their TBARS levels. The animals treated with metformin or bixin-rich annatto extract, alone or in combination, showed significant decreases in the levels of TBARSs in their livers and kidneys. A similar result was found by Ma et al. [9], with a reduction in the levels of malondialdehyde, a biomarker of lipoperoxidation, and an increase in the activities of antioxidant enzymes in the kidneys of the mice through a mechanism dependent on NRF-2 activation in a CCl4-induced kidney inflammation model. Bixin and other carotenoids have important mechanisms of action in regulating inflammatory responses, modulating NF-kB activity and adipocyte differentiation. Carotenoids are involved in the activation of important components, including PPARγ, which is related to the activation of lipogenesis pathways [42]. The activation of PPARγ is fundamental to the differentiation of adipocytes, increasing the hydrolysis rates of the triglycerides present in lipoproteins via the activation of lipoprotein lipase; concomitantly, PPARγ promotes an increase in the transcription of genes involved in the uptake of fatty acids, including the fatty acid transport protein (FATP) and the fatty acid translocase (FAT/CD36) [43]. 

The activation of PPARγ can lead to a recovery in the lipid storage capacity of the adipose tissues via preservation of adipocytic function, contributing to increased insulin sensitivity [44]. In addition, the present study showed the antiobesogenic potential of both metformin and bixin-rich annatto extract, whether administered alone or in combination. In parallel, these treatments caused improvements in insulin sensitivity. These improvements may be related, at least in part, to the effects of bixin on PPARγ activation in the white adipose tissues. Since bixin is capable of activating PPARγ, its beneficial effects related to antiobesogenic potential and insulin sensitivity restoration may be a consequence of improvements in adipocyte function in a pathway that is not dependent on the dose administered to the animals.

## 4. Materials and Methods

### 4.1. Preparation of the Nanostructured Lipid System 

The nanostructured lipid system had the following composition: oil phase: sunflower oil (5%); surfactant mixture: Brij O20/soy phosphatidylcholine—2:1 (10%); and aqueous phase: phosphate buffer (pH of 7.4) (85%). Bixin-rich annatto extract and metformin, alone or in combination, were incorporated into the oil phase and the surfactant mixture. Next, the aqueous phase was added, and the mixtures were sonicated with a rod sonicator (Q500 of QSonica^®®^, Newtown, CT, USA) in an ice bath (25 min at 30 s intervals every minute). The formulations containing the bioactives were centrifuged (11,180× *g* for 15 min) [45].

The formulations based on the nanostructured lipid system contained 1.1 or 2.2 mg/mL bixin-rich annatto extract and/or 10 mg/mL metformin, allowing them to reach final doses of 5.5 or 11 mg/kg bixin-rich annatto extract and/or 50 mg/kg metformin, respectively, alone or in combination. 

### 4.2. Animal Model—Induction of Obesity/Insulin Resistance and Experiment Conclusion

C57BL-6J male mice with body weight values of 21 ± 0.17 g (4 weeks old) were obtained from Anilab (Animais de Laboratório, Criação e Comércio LTDA, Paulínia, Brazil). The mice were housed in polypropylene cages (two animals per cage) and maintained under a light/dark cycle (12 h) and controlled conditions of humidity (55 ± 5%) and temperature (23 ± 1 °C) at the Bioterium of the Department of Clinical Analysis (FCF/UNESP). During the acclimation period (2 weeks), a standard chow diet (Presence Nutrição Animal, Paulínia, São Paulo, Brazil) and water were offered to the mice ad libitum. Next, the mice were fed with a control diet (C; 3.85 kcal/g; 4% lipids) or with a high-fat diet (HFD; 5.40 kcal/g; 35% lipids) (Pragsoluções Biociências Serviços e Comércio Ltd., Jaú, Brazil) (Appendix A). The HFD enabled the development of obesity and insulin resistance.

The mice were divided into eight groups (*n* = 12 per group; total of 96 mice): the C group (non-obese mice fed a C diet), the H group (obese, HFD-fed mice), the V group (HFD-fed mice treated with a vehicle—the nanostructured lipid system), the M group (HFD-fed mice treated with 50 mg/kg metformin), the B1 group (HFD-fed mice treated with 5.5 mg/kg bixin-rich annatto extract), the MB1 group (HFD-fed mice treated with 50 mg/kg metformin + 5.5 mg/kg bixin-rich annatto extract), the B2 group (HFD-fed mice treated with 11 mg/kg bixin-rich annatto extract), and the MB2 group (HFD-fed mice treated with 50 mg/kg metformin + 11 mg/kg bixin-rich annatto extract). To allocate the animals into the experimental groups, stratified randomization was used. In addition, standardization of the median body weight values across the groups was applied to composing the experimental groups containing the HFD-fed mice. The inclusion criteria for the groups containing the HFD-fed mice were the animals having body weight values of approximately 30 g before the beginning of the treatments. The exclusion criteria for groups containing the HFD-fed mice were low body weight gain values.

All the treatments were administered orogastrically (gavage). Annato extract powder rich in bixin (*Bixa orellana* L., 60% bixin, Lichnoflora Pesquisa e Desenvolvimento em Produtos Naturais Ltd., Ribeirão Preto, Brazil) and metformin (99.56% metformin hydrochloride, Abhilash Chemicals and Pharmaceuticals, Madurai, India) were incorporated into a nanostructured lipid system. The doses of bixin-rich annatto extract (5.5 and 11 mg/kg) were chosen based on our previous study [36], and the dose of metformin (50 mg/kg) was based on studies performed in mice under experimental models of obesity/insulin resistance and treated with metformin [28,29], and we reduced the dose of the drug since it was administered in a nanostructured lipid system. The experiment was performed for 17 weeks; the mice were treated for the last 8 weeks via daily gavage (5 μL/g animal) between 08:00 and 09:00 h from the 9th to the 17th weeks. The mice from the V group received the vehicle (the nanostructured lipid system without metformin or bixin-rich annatto extract) via daily gavage (5 μL/g animal). Finally, to mimic the oral treatments, the mice from the C and H groups received water by gavage.

Their body weight and food intake were monitored weekly throughout the experimental period. By multiplying their food intake values (g) by the energy values (kcal) provided for the diets, the energy intake values were obtained. The oral glucose tolerance test (OGTT) was performed at week 15 (which corresponded to 6 weeks of treatments). In 12 h-fasted mice, the OGTT was performed at 11:00 a.m. after gavage with 1 g/kg glucose (glucose overload). Their glycemia levels were monitored before (0 min) and after (15, 30, 60, 90, and 120 min) the glucose overload. A drop of peripheral blood was obtained from the mice’s tails to determine their glycemia levels, which were measured by a glucometer (Abbott Diabetes Care Ltd., São Paulo, Brazil) [46]. The insulin tolerance test (ITT) was performed at week 16 (which corresponded to 7 weeks of treatments). In 2 h-fasted mice, the ITT was performed at 01:00 p.m. with an intraperitoneal injection of insulin (0.4 UI/kg). Their glycemia levels were monitored before (0 min) and after (15, 30, 45, and 60 min) the administration of insulin [46].

At week 17 (which corresponded to 8 weeks of treatments), the mice were fasted for 6 h and anesthetized by intraperitoneal administration of xylazine–ketamine (16 mg/kg xylazine–90 mg/kg ketamine). Next, the mice were euthanized by exsanguination via cardiac puncture under anesthesia, and blood samples were collected into heparinized tubes and then centrifuged (700× *g* for 10 min at 25 °C) to obtain plasma samples. Their livers, kidneys, hearts, gastrocnemius skeletal muscles, epididymal white adipose tissues, and interscapular brown adipose tissues were removed, weighed, snap-frozen in liquid nitrogen, and frozen (−80 °C).

All the experimental procedures were previously approved by the Committee for Ethics in Animal Experimentation at the School of Pharmaceutical Sciences, UNESP (CEUA/FCF/CAr n^o^ 19/2019).

### 4.3. Analysis of Plasma Biochemical Markers

Their plasma levels of glucose, total cholesterol, high-density lipoprotein cholesterol (HDL-cholesterol), triglycerides, albumin, alanine aminotransferase (ALT), creatinine, and uric acid were measured using commercial kits (Labtest Diagnostica SA, Lagoa Santa, Brazil).

### 4.4. Analysis of Glycoxidative Stress Biomarkers

In the plasma and livers, fluorescent AGEs were measured using 1.2 M chloroform, 0.12 M trichloroacetic acid, and 0.1 M sodium hydroxide (plasma) or 2.4 M chloroform (liver homogenates) [47]. In the kidney homogenates, fluorescent AGEs were measured using 0.1 M sodium hydroxide [48]. Next, the tubes containing these samples were vigorously shaken, and then the tubes were maintained at 10 ± 2 °C for 30 min, followed by centrifugation (10,000× *g* for 10 min at 10 °C). Tissue supernatants were used to measure the fluorescence intensity relative to the AGEs, with excitation and emission wavelengths of 370 nm and 440 nm, respectively, in a microplate multi-mode reader with the split set at 16 nm (Synergy H1TM, BioTek Instruments Inc., Winooski, VT, USA). The results were expressed as arbitrary units (AU) of fluorescence per milligram of protein.

Deproteinized tissue samples (plasma, livers, and kidneys) were used to determine the lipid peroxidation products via a thiobarbituric acid (TBA) assay [49]. The thiobarbituric acid-reactive substances (TBARSs), including malondialdehyde, were measured via spectrofluorescence (with excitation and emission wavelengths of 510 nm and 553 nm, respectively) in the plasma and via spectrophotometry (535 nm) in the livers and kidneys. The results were expressed in terms of μmol/L (plasma) and μmol/g tissue (livers and kidneys).

### 4.5. Analysis of Endogenous Antioxidant Defenses

The livers and kidneys were used to determine the activities of superoxide dismutase (SOD), catalase (CAT), and glutathione peroxidase (GSH-Px), and the plasma samples were used to determine the activity of paraoxonase 1 (PON-1), according to standardized methods. 

Sample preparations: The livers and kidneys (0.1 g) were prepared in 1 mL sodium phosphate buffer (10 mmol/L, pH 7.4) at 4 °C. The homogenates were then centrifuged at 10,000× *g* for 10 min at 4 °C. The activities of SOD, CAT, and GSH-Px were measured in the liver and kidney supernatants.

The SOD activity was measured by monitoring the inhibition of the reduction of nitroblue tetrazolium (NBT). Xanthine oxidase catalyzes xanthine’s oxidation, and superoxide anion radical (O_2_^•−^) molecules are produced, which reduce NBT into a formazan. The SOD present in the sample catalyzes the dismutation of O_2_^•−^, inhibiting the reduction of NBT, which is monitored at 550 nm. The results were expressed in terms of U/mg protein. One SOD unit is defined as the amount of the enzyme required to inhibit the rate of NBT reduction by 50% [50].

The CAT activity was monitored by the consumption of hydrogen peroxide (H_2_O_2_) at 230 nm. The results were expressed in terms of μmol of H_2_O_2_ consumed/min/mg protein [51].

The GSH-Px activity was determined by monitoring the oxidation of the reduced form of nicotinamide adenine dinucleotide phosphate (NADPH) into NADP+. GSH-Px catalyzes the oxidation of the reduced form of glutathione (GSH) in the presence of H_2_O_2_. Glutathione in its oxidized form (GSSG) is reduced into GSH by glutathione reductase (GSH-Rd), with the concomitant oxidation of NADPH into NADP+, monitored at 340 nm. The results were expressed in terms of μmol of NADPH oxidized/min/mg protein [52].

The activity of PON-1 in the plasma was measured by monitoring p-nitrophenol at 405 nm, which is released after paraoxon’s hydrolysis by PON-1 [36]. The results were expressed in terms of U/mg HDL-cholesterol (unit = μmoL paraoxon hydrolyzed/min).

The protein levels in the tissue samples were determined according to Lowry et al. [53] to correct the results for SOD, CAT, GSH-Px, and fluorescent AGEs.

### 4.6. Statistical Analysis

The data are expressed as mean ± standard error of mean (SEM). To compare the intergroup differences, one-way analysis of variance followed by the Student–Newman–Keuls test was used. To compare the intragroup changes in the body weight values relative to week 0, a paired Student’s t-test was used. The data were considered statistically different at *p* < 0.05 (*), *p* < 0.01 (**), and *p* < 0.001 (***). Statistical analyses were performed using GraphPad Prism 6.01 (GraphPad Software, San Diego, CA, USA).

## 5. Conclusions

Combining current medicines with bioactives of a natural origin might be an interesting therapeutic strategy to attenuate or even prevent the complications occurring in obesity and diabetes mellitus. In this regard, the present study showed that metformin combined with bixin-rich annatto extract had antiobesogenic effects and decreased the cholesterol levels and improved the insulin sensitivity in mice fed a high-fat diet, which represents the maintenance of metformin’s effects and bixin’s effects when they are administered alone. In addition, metformin + bixin-rich annatto extract triggered cytoprotective mechanisms that counteracted the harmful impacts of oxidative stress by increasing their endogenous antioxidant defenses. To the best of our knowledge, by targeting not only obesity, hyperglycemia, and insulin resistance but also hypercholesterolemia and oxidative stress, this study provides the first evidence of a therapeutic strategy based on the combination of bixin and metformin for combating the complications of metabolic disturbances resulting from oxidative stress.

## Figures and Tables

**Figure 1 pharmaceuticals-17-01202-f001:**
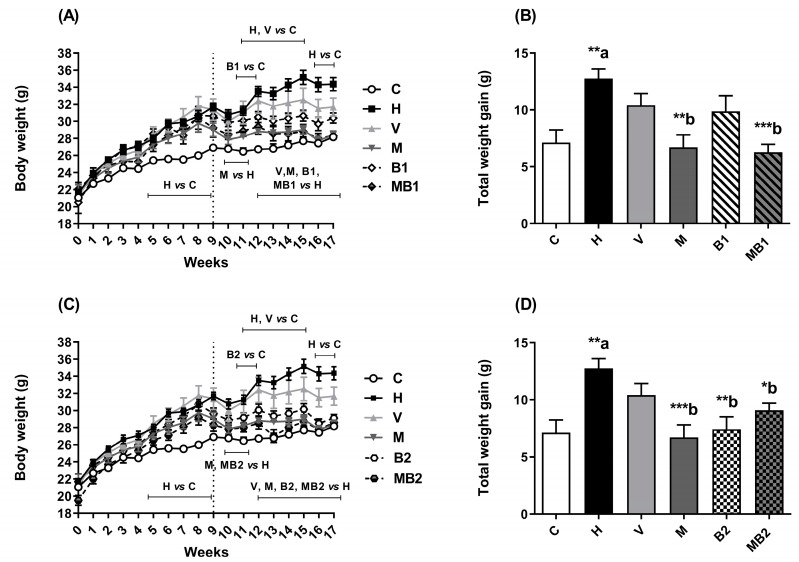
Body weight of HFD-fed mice treated for 8 weeks with metformin and/or 5.5 and 11 mg/kg bixin-rich annatto extract. Evolution of body weight (**A**,**B**) and total weight gain from week 0 to week 17 (**C**,**D**). Results are expressed as mean ± standard error. C: mice fed control diet; H: mice fed HFD; V: mice fed HFD and treated with vehicle; M: mice fed HFD and treated with 50 mg/kg metformin; B1: mice fed HFD and treated with 5.5 mg/kg bixin-rich extract; MB1: mice fed HFD and treated with 50 mg/kg metformin + 5.5 mg/kg bixin-rich extract; B2: mice fed HFD and treated with 11 mg/kg bixin-rich extract; MB2: mice fed HFD and treated with 50 mg/kg metformin + 11 mg/kg bixin-rich extract. Differences between groups were considered significant at * *p* < 0.05, ** *p* < 0.01, and *** *p* < 0.001. a, differences from C; b, differences from H.

**Figure 2 pharmaceuticals-17-01202-f002:**
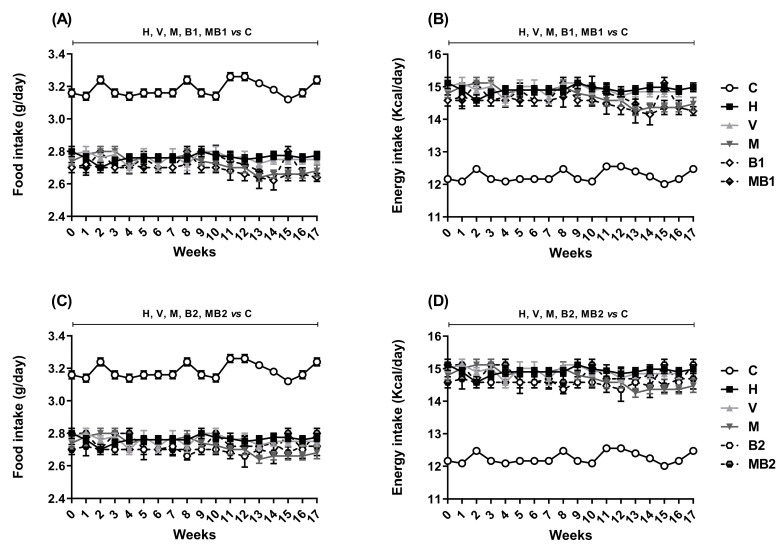
Food and energy intake of HFD-fed mice treated for 8 weeks with metformin and/or 5.5 and 11 mg/kg bixin-rich annatto extract. Food intake (**A**,**C**) and energy intake (**B**,**D**). Results are expressed as mean ± standard error. C: mice fed control diet; H: mice fed HFD; V: mice fed HFD and treated with vehicle; M: mice fed HFD and treated with 50 mg/kg metformin; B1: mice fed HFD and treated with 5.5 mg/kg bixin-rich extract; MB1: mice fed HFD and treated with 50 mg/kg metformin + 5.5 mg/kg bixin-rich extract; B2: mice fed HFD and treated with 11 mg/kg bixin-rich extract; MB2: mice fed HFD and treated with 50 mg/kg metformin + 11 mg/kg bixin-rich extract. Differences between groups were considered significant at *p* < 0.05.

**Figure 3 pharmaceuticals-17-01202-f003:**
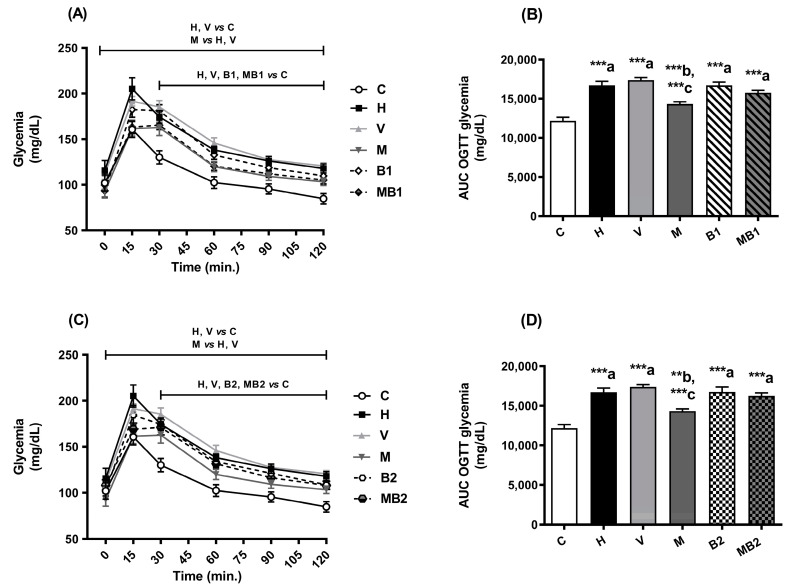
Glucose tolerance of HFD-fed mice treated for 8 weeks with metformin and/or 5.5 and 11 mg/kg bixin-rich annatto extract. Oral glucose tolerance test (OGTT) (**A**,**C**) and AUC of OGTT (**B**,**D**). Results are expressed as mean ± standard error. AUC: area under the curve; C: mice fed control diet; H: mice fed HFD; V: mice fed HFD and treated with vehicle; M: mice fed HFD and treated with 50 mg/kg metformin; B1: mice fed HFD and treated with 5.5 mg/kg bixin-rich extract; MB1: mice fed HFD and treated with 50 mg/kg metformin + 5.5 mg/kg bixin-rich extract; B2: mice fed HFD and treated with 11 mg/kg bixin-rich extract; MB2: mice fed HFD and treated with 50 mg/kg metformin + 11 mg/kg bixin-rich extract. Differences between groups were considered significant at ** *p* < 0.01 and *** *p* < 0.001. a, differences from C; b, differences from H; c, differences from V.

**Figure 4 pharmaceuticals-17-01202-f004:**
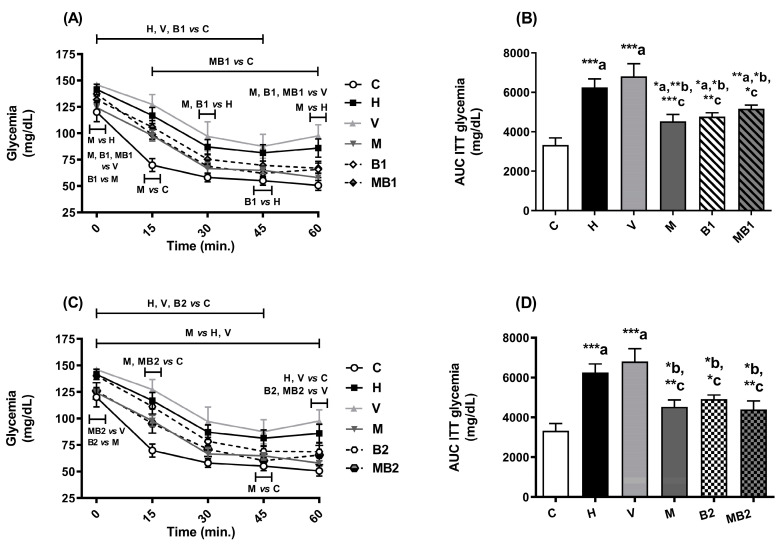
Insulin sensitivity of HFD-fed mice treated for 8 weeks with metformin and/or 5.5 and 11 mg/kg bixin-rich annatto extract. Insulin tolerance test (ITT) (**A**,**C**) and AUC of ITT (**B**,**D**). Results are expressed as mean ± standard error. AUC: area under the curve; C: mice fed control diet; H: mice fed HFD; V: mice fed HFD and treated with vehicle; M: mice fed HFD and treated with 50 mg/kg metformin; B1: mice fed HFD and treated with 5.5 mg/kg bixin-rich extract; MB1: mice fed HFD and treated with 50 mg/kg metformin + 5.5 mg/kg bixin-rich extract; B2: mice fed HFD and treated with 11 mg/kg bixin-rich extract; MB2: mice fed HFD and treated with 50 mg/kg metformin + 11 mg/kg bixin-rich extract. Differences between groups were considered significant at * *p* < 0.05, ** *p* < 0.01, and *** *p* < 0.001. a, differences from P; b, differences from H; c, differences from V.

**Figure 5 pharmaceuticals-17-01202-f005:**
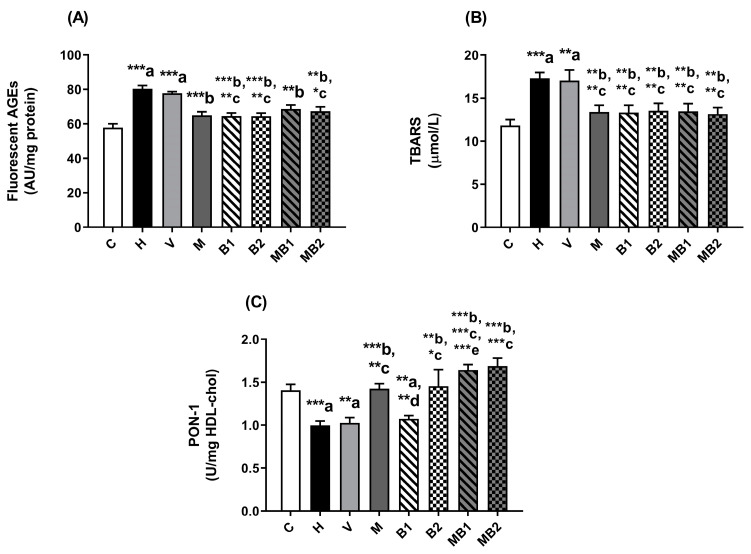
Biomarkers of glycoxidative damage and antioxidant defenses in plasma of HFD-fed mice treated for 8 weeks with metformin and/or 5.5 and 11 mg/kg bixin-rich annatto extract. Fluorescent AGEs (**A**), TBARSs (**B**), and activity of PON-1 (**C**). Results are expressed as mean ± standard error. C: mice fed control diet; H: mice fed HFD; V: mice fed HFD and treated with vehicle; M: mice fed HFD and treated with 50 mg/kg metformin; B1: mice fed HFD and treated with 5.5 mg/kg bixin-rich extract; MB1: mice fed HFD and treated with 50 mg/kg metformin + 5.5 mg/kg bixin-rich extract; B2: mice fed HFD and treated with 11 mg/kg bixin-rich extract; MB2: mice fed HFD and treated with 50 mg/kg metformin + 11 mg/kg bixin-rich extract. Differences between groups were considered significant at * *p* < 0.05, ** *p* < 0.01, and *** *p* < 0.001. a, differences from P; b, differences from H; c, differences from V; d, differences from M; e, differences from B1.

**Figure 6 pharmaceuticals-17-01202-f006:**
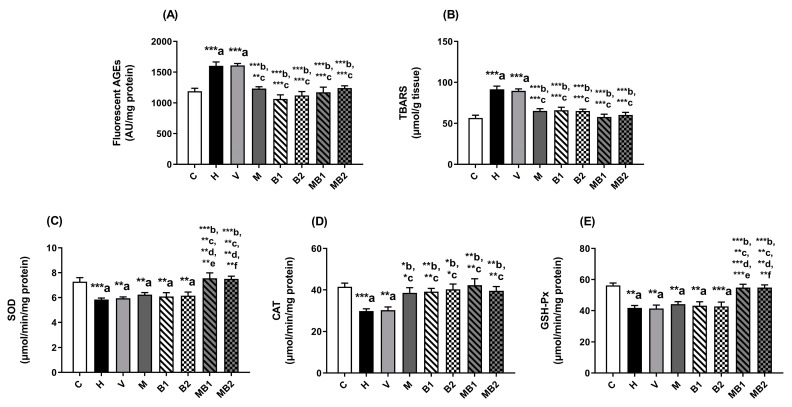
Biomarkers of glycoxidative damage and antioxidant defenses in livers of HFD-fed mice treated for 8 weeks with metformin and/or 5.5 and 11 mg/kg bixin-rich annatto extract. Fluorescent AGEs (**A**), TBARSs (**B**), and activities of SOD (**C**), CAT (**D**), and GSH-Px (**E**). Results are expressed as mean ± standard error. C: mice fed control diet; H: mice fed HFD; V: mice fed HFD and treated with vehicle; M: mice fed HFD and treated with 50 mg/kg metformin; B1: mice fed HFD and treated with 5.5 mg/kg bixin-rich extract; MB1: mice fed HFD and treated with 50 mg/kg metformin + 5.5 mg/kg bixin-rich extract; B2: mice fed HFD and treated with 11 mg/kg bixin-rich extract; MB2: mice fed HFD and treated with 50 mg/kg metformin + 11 mg/kg bixin-rich extract. Differences between groups were considered significant at * *p* < 0.05, ** *p* < 0.01 and *** *p* < 0.001. a, differences from P; b, differences from H; c, differences from V; d, differences from M; e, differences from B1; f, differences from B2.

**Figure 7 pharmaceuticals-17-01202-f007:**
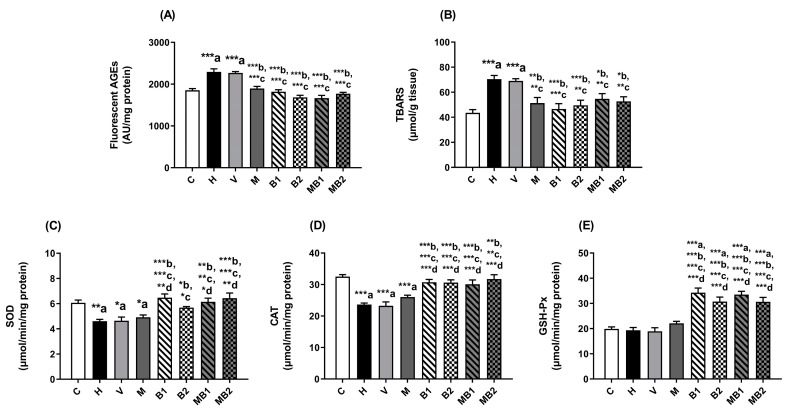
Biomarkers of glycoxidative damage and antioxidant defenses in kidneys of HFD-fed mice treated for 8 weeks with metformin and/or 5.5 and 11 mg/kg bixin-rich annatto extract. Fluorescent AGEs (**A**), TBARSs (**B**), and activities of SOD (**C**), CAT (**D**), and GSH-Px (**E**). Results are expressed as mean ± standard error. C: mice fed control diet; H: mice fed HFD; V: mice fed HFD and treated with vehicle; M: mice fed HFD and treated with 50 mg/kg metformin; B1: mice fed HFD and treated with 5.5 mg/kg bixin-rich extract; MB1: mice fed HFD and treated with 50 mg/kg metformin + 5.5 mg/kg bixin-rich extract; B2: mice fed HFD and treated with 11 mg/kg bixin-rich extract; MB2: mice fed HFD and treated with 50 mg/kg metformin + 11 mg/kg bixin-rich extract. Differences between groups were considered significant at * *p* < 0.05, ** *p* < 0.01, and *** *p* < 0.001. a, differences from P; b, differences from H; c, differences from V; d, differences from M.

**Table 1 pharmaceuticals-17-01202-t001:** Weights of tissues (mg/mm of tibia) of HFD-fed mice treated for 8 weeks with metformin and/or 5.5 and 11 mg/kg bixin-rich annatto extract.

Groups	C	H	V	M	B1	B2	MB1	MB2
Liver	59.5 ± 1.0	64.1 ± 1.9	58.3 ± 2.2	57.4 ± 1.8	57.2 ± 1.6	57.5 ± 1.8	55.0 ± 1.8 ^b^	57.9 ± 2.2
Kidneys	9.2 ± 0.6	10.4 ± 0.7	9.7 ± 0.7	9.1 ± 0.6	9.4 ± 0.8	9.7 ± 0.7	9.1 ± 0.4	6.7 ± 0.3
Heart	7.0 ± 0.2	7.4 ± 0.1	7.1 ± 0.2	7.1 ± 0.2	7.3 ± 0.2	7.1 ± 0.2	7.1 ± 0.2	7.1 ± 0.2
Muscles (gastrocnemius)	8.1 ± 0.1	8.4 ± 0.2	8.1 ± 0.1	8.1 ± 0.1	8.1 ± 0.1	8.2 ± 0.1	8.1 ± 0.1	8.1 ± 0.1
WAT (epididymal)	26.7 ± 1.5	66.4 ± 5.3 ^a^	54.6 ± 3.8 ^a,b^	41.2 ± 5.0 ^b^	37.4 ± 2.9 ^b^	43.3 ± 4.0 ^b^	44.2 ± 4.8 ^a,b^	42.1 ± 3.2 ^b^
BAT (interscapular)	4.1 ± 0.2	3.8 ± 0.3	3.5 ± 0.2	3.0 ± 0.2 ^a^	2.9 ± 0.1 ^a^	3.2 ± 0.2 ^a^	3.2 ± 0.2 ^a^	3.1 ± 0.2 ^a^

Results are expressed as mean ± standard error. C: mice fed control diet; H: mice fed HFD; V: mice fed HFD and treated with vehicle; M: mice fed HFD and treated with 50 mg/kg metformin; B1: mice fed HFD and treated with 5.5 mg/kg bixin-rich extract; MB1: mice fed HFD and treated with 50 mg/kg metformin + 5.5 mg/kg bixin-rich extract; B2: mice fed HFD and treated with 11 mg/kg bixin-rich extract; MB2: mice fed HFD and treated with 50 mg/kg metformin + 11 mg/kg bixin-rich extract. Differences between groups were considered significant at *p* < 0.05. a, differences from C; b, differences from H.

**Table 2 pharmaceuticals-17-01202-t002:** Plasma levels of biochemical markers in plasma of HFD-fed mice treated for 8 weeks with metformin and/or 5.5 and 11 mg/kg bixin-rich annatto extract.

Groups	C	H	V	M	B1	B2	MB1	MB2
Triglycerides (mg/dL)	53.4 ± 7.1	41.3 ± 8.4 ^a^	41.6 ± 5.5 ^a^	38.6 ± 10.1 ^a^	38.1 ± 8.0 ^a^	39.8 ± 5.3 ^a^	39.4 ± 10.5 ^a^	39.5 ± 6.2 ^a^
Total Cholesterol (mg/dL)	98.3 ± 3.3	135.6 ± 15.8 ^a^	131.2 ± 19.4 ^a^	107.3 ± 14.1 ^b^	113.3 ± 9.6 ^b^	112.3 ± 14.1 ^b^	108.5 ± 8.7 ^b^	116.0 ± 9.4 ^b^
HDL-Cholesterol (mg/dL)	79.3 ± 2.5	108.3 ± 15.1 ^a^	99.8 ± 7.5 ^a^	89.7 ± 14.4 ^b^	94.2 ± 2.8	94.0 ± 8.3	90.1 ± 6.6 ^b^	90.8 ± 12.2 ^b^
Creatinine (mg/dL)	0.16 ± 0.003	0.15 ± 0.002	0.16 ± 0.004	0.15 ± 0.003	0.16 ± 0.011	0.16 ± 0.003	0.16 ± 0.003	0.15 ± 0.003
Uric Acid (mg/dL)	0.4 ± 0.02	0.4 ± 0.02	0.3 ± 0.03	0.3 ± 0.03	0.4 ± 0.09	0.4 ± 0.04	0.4 ± 0.04	0.4 ± 0.07
Albumin (g/dL)	2.1 ± 0.02	2.1 ± 0.03	2.1 ± 0.03	2.1 ± 0.04	2.1 ± 0.03	2.1 ± 0.02	2.1 ± 0.03	2.1 ± 0.03
ALT (U/L)	24.8 ± 2.3	28.4 ± 4.4	27.3 ± 2.4	21.4 ± 2.4	24.4 ± 1.6	21.5 ± 3.1	24.1 ± 2.5	23.0 ± 4.0

Results are expressed as mean ± standard error. C: mice fed control diet; H: mice fed HFD; V: mice fed HFD and treated with vehicle; M: mice fed HFD and treated with 50 mg/kg metformin; B1: mice fed HFD and treated with 5.5 mg/kg bixin-rich extract; MB1: mice fed HFD and treated with 50 mg/kg metformin + 5.5 mg/kg bixin-rich extract; B2: mice fed HFD and treated with 11 mg/kg bixin-rich extract; MB2: mice fed HFD and treated with 50 mg/kg metformin + 11 mg/kg bixin-rich extract. Differences between groups were considered significant at *p* < 0.05. a, differences from C; b, differences from H.

## Data Availability

The datasets generated during and/or analyzed during the current study are available from the corresponding author on reasonable request.

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
