# Peer review of "Bixin Combined with Metformin Ameliorates Insulin Resistance and Antioxidant Defenses in Obese Mice"

_pharmaceuticals, 2024, doi:10.3390/ph17091202_

Round 1

Reviewer 1 Report

Comments and Suggestions for Authors

This article focused on the improvement of insulin resistance and antioxidant potential in obese mice model. This manuscript can be very interesting for the scientific community. This paper still lacks of few anomalies. Please follow the instructions:

1. In the abstract you should write a line or two about bixin.

2. Result is not nicely focused in abstract section. Please rewrite it.

3. Bixin is again less focused in introduction section.

4. In every statistical analysis, you consider significance based on p < 0.05. please revise it as ***p<0.001, **p< 0.01, *p< 0.05

5. Please check the alpha-amylase and alpha-glucosidase, DPP-4 inhibitory action, GLP-1, HBA1c for concrete idea on antidiabetic potential .

5. Mention the justification for dosing of sample/standard.

6. To evaluate the antioxidant property, check DPPH free radical scavenging, total phenolic and flavonoid control, as well as gallic acid equivalency protocol. 

7. The discussion is well written. But provide some scientific evidence to link between result and bixin.

8. How the animals were sacrificed or what was the fate? Please mention it in 4.2. Animal Experimental Design section.

9. COnclusion is very poorly written. rewrite it.

Comments on the Quality of English Language

A minor correction is recommended.

Reviewer 2 Report

Comments and Suggestions for Authors

This study investigated the effects of combining metformin with an extract from the fruit seed of Bixa orellana L., which rich in bixin, on obesity-induced complications in mice fed a high-fat diet (HFD). The research specifically examines the combination's ability to reduce weight gain, improve insulin sensitivity, lower total cholesterol, and enhance endogenous antioxidant defenses. Overall, the experimental design and results of this study are highly interesting and innovative, though there are some areas that I suggest for revision.

Specific comments:

1.    Figure 1A and 1C: The V, M, B1, MB1, B2, and MB2 groups in the figures should had been fed a HFD from week 0 to week 9, under the same conditions as the H group, and then with different treatments starting from week 9 until the experiment concluded at week 17. Why did the mice in these groups have lower body weights compared to the H group during weeks 0 to 9?

2.  Figure 1B and 1D: Please clarify in the figure legend whether the total weight gain is calculated from week 0 to week 17 or from week 9 to week 17. Given that the body weights of the other groups were lower than those of the H group during weeks 0 and 9, the calculation period for total weight gain becomes crucial.

3.   Figure 1B and 1D: The data show that the V group exhibited weight loss effects. It is recommended to provide an explanation for this observation in the discussion section.

4.     Table 1: For the Liver category, please recheck the significance of the data for the MB1 group; it seems that should be marked with “b”. For the BAT category, the M, B1, MB1, B2, and MB2 groups seem to warrant a marking ofa and/or b.

5.  Figure 1B and 1D: Please explain in the text what "AUC" on the y-axis stands for.

6.     Line 14: I suggest changing "bixin-rich extract" to "bixin-rich Bixa orellana fruit seed extract" or "bixin-rich annatto extract," so that readers can immediately understand the source of the bixin used in the study, preventing confusion with a pure compound.

7.  Line 78: In scientific writing, especially at the beginning of the results section, using uncertain language such as “It is possible to observe” is inappropriate. I recommend using more definitive wording.

8.     Line 216: Should the word “addictive” be “additive”?

9. The manuscript contains many abbreviations that are difficult to understand due to the lack of corresponding full terms in the text. I suggest the authors provide a list of abbreviations.

Round 2

Reviewer 1 Report

Comments and Suggestions for Authors

It can be accepted now

Comments on the Quality of English Language

This is finely tuned now